Homeoprotein OTX1 and OTX2 involvement in rat myenteric neuron adaptation after DNBS-induced colitis

Bistoletti Michela 1
Micheloni Giovanni 1
Baranzini Nicolò 2
Bosi Annalisa 1
Conti Andrea 1
Filpa Viviana 1
Pirrone Cristina 1
Millefanti Giorgia 1
Moro Elisabetta 3
Grimaldi Annalisa 2
Valli Roberto 1
http://orcid.org/0000-0003-0088-2712 Baj Andreina 1
Crema Francesca 3
Giaroni Cristina 1 cristina.giaroni@uninsubria.it
Porta Giovanni 1
1 Department of Medicine and Surgery, University of Insubria , Varese , Italy
2 Department of Biotechnology and Life Sciences, University of Insubria , Varese , Italy
3 Department of Internal Medicine and Therapeutics, University of Pavia , Pavia , Italy
Silva Jerson
Electronic publication date: 2020 Feb 13
Publication date: 2020
Volume: 8
Electronic Location ID: e8442
Received 2019 Sep 13; Accepted 2019 Dec 20
Copyright: © 2020 Bistoletti et al.
Copyright year: 2020
Copyright holder: Bistoletti et al.
License: This is an open access article distributed under the terms of the Creative Commons Attribution License, which permits unrestricted use, distribution, reproduction and adaptation in any medium and for any purpose provided that it is properly attributed. For attribution, the original author(s), title, publication source (PeerJ) and either DOI or URL of the article must be cited.
License URL: https://creativecommons.org/licenses/by/4.0/

Keywords: Myenteric plexus, Inflammation, Plasticity, Homeobox genes, OTX1, OTX2, iNOS, nNOS

Funding: University of Insubria FAR 2016-2018 University of Pavia FAR 2016-2018 This study was supported by the University of Insubria (FAR 2016-2018 to Cristina Giaroni, Andreina Baj, Annalisa Grimaldi) and the University of Pavia (FAR 2016-2018 to Francesca Crema). The funders had no role in study design, data collection and analysis, decision to publish, or preparation of the manuscript.

==============================
Background

Inflammatory bowel diseases are associated with remodeling of neuronal circuitries within the enteric nervous system, occurring also at sites distant from the acute site of inflammation and underlying disturbed intestinal functions. Homeoproteins orthodenticle OTX1 and OTX2 are neuronal transcription factors participating to adaptation during inflammation and underlying tumor growth both in the central nervous system and in the periphery. In this study, we evaluated OTX1 and OTX2 expression in the rat small intestine and distal colon myenteric plexus after intrarectal dinitro-benzene sulfonic (DNBS) acid-induced colitis.

Methods

OTX1 and OTX2 distribution was immunohistochemically investigated in longitudinal muscle myenteric plexus (LMMP)-whole mount preparations. mRNAs and protein levels of both OTX1 and OTX2 were evaluated by qRT-PCR and Western blotting in LMMPs.

Results

DNBS-treatment induced major gross morphology and histological alterations in the distal colon, while the number of myenteric neurons was significantly reduced both in the small intestine and colon. mRNA levels of the inflammatory markers, TNFα, pro-IL1β, IL6, HIF1α and VEGFα and myeloperoxidase activity raised in both regions. In both small intestine and colon, an anti-OTX1 antibody labeled a small percentage of myenteric neurons, and prevalently enteric glial cells, as evidenced by co-staining with the glial marker S100β. OTX2 immunoreactivity was present only in myenteric neurons and was highly co-localized with neuronal nitric oxide synthase. Both in the small intestine and distal colon, the number of OTX1- and OTX2-immunoreactive myenteric neurons significantly increased after DNBS treatment. In these conditions, OTX1 immunostaining was highly superimposable with inducible nitric oxide synthase in both regions. OTX1 and OTX2 mRNA and protein levels significantly enhanced in LMMP preparations of both regions after DNBS treatment.

Conclusions

These data suggest that colitis up-regulates OTX1 and OTX2 in myenteric plexus both on site and distantly from the injury, potentially participating to inflammatory-related myenteric ganglia remodeling processes involving nitrergic transmission.

Introduction

Inflammatory bowel diseases (IBD) are chronic inflammatory disorders of the gastrointestinal tract with increasing incidence worldwide (Ng et al., 2018). Inflammation develops because of an exaggerated immune response to luminal antigens derived from the gut microbiota or from infecting pathogens in genetically predisposed individuals (Ni et al., 2017; Baj et al., 2019a). Genetic and environmental factors also contribute to IBD development, although the exact atiology remains to be fully disclosed. Inflammation leads to profound alterations of the intestinal architecture, which influence intrinsic neuronal circuitries constituting the enteric nervous system (ENS). Neuronal adaptation includes morphology, excitability and synaptic changes, with consequent alterations of sensory, motor and secretory functions contributing to the development of IBD-associated symptoms (Brierley & Linden, 2014; De Schepper et al., 2008; Baj et al., 2019b). The cross-talk occurring among different cell populations of the enteric microenvironment (i.e., neurons, enteric glia, interstitial cells of Cajal, immunocytes), the saprophytic microbial flora and infiltrating inflammatory cells may account for the structural and functional changes occurring in enteric circuitries in different physiological and pathological conditions, including inflammation (Giaroni, 2015; Lomax, Fernández & Sharkey, 2005; Bistoletti et al., 2019). For example, neuro-immune interactions may help explain the occurrence of damage at non-inflamed gastrointestinal sites, distant from the inflammatory process (Brierley & Linden, 2014; De Schepper et al., 2008). It is, thus, particularly important to unveil possible molecular pathways sustaining neuronal degeneration during inflammatory states, both on site and, more distantly, along the gastrointestinal tract. Indeed, research in this field opens an exciting potential scenario where new molecules displaying high efficacy in the treatment of IBD may became available in association and/or substitution of conventional anti-inflammatory, immunosuppressive and biologic drugs (Pagano et al., 2016; Neurath, 2017; Szebeni et al., 2019; Pagano et al., 2019).

In the present study, we investigated possible changes in the expression of homeoprotein transcription factors, orthodenticle homeobox protein 1 and 2 (OTX1 and OTX2), in the rat myenteric plexus after an experimentally-induced colitis. OTX1 and OTX2 are transcription factors derived from homeobox-containing genes located, in humans, on chromosome 2p15 and 14q22, respectively (Kastury et al., 1994). OTX1 and OTX2 homeobox genes are the vertebrate orthologues to the Drosophila orthodenticle homeobox genes and are essential for the specification, regionalization and terminal differentiation of the rostral part of the central nervous system during development (Acampora et al., 2001; Larsen et al., 2010). OTX2 is an essential factor influencing the maturation of various neuronal subpopulations, including oculomotor, thalamic glutamatergic progenitors and midbrain dopaminergic neurons (Sherf et al., 2015; Puelles et al., 2006). Recent evidence suggest for OTX2 a fundamental role in regulating neuronal plasticity in the visual cortex (Bernard & Prochiantz, 2016). During development in both rodent and human, OTX1 is prevalently expressed in proliferative zones of the neocortex and is described to have a fundamental role in determining the volume and proportion of the visual cortex (Ando et al., 2008). Despite their central role in neuronal development and maturation, there are lines of evidence suggesting that both OTX1 and OTX2 are expressed also in adult tissues and may participate to the development of pathological conditions linked to inflammation (Housset et al., 2013; Azzolini et al., 2013). In addition, the involvement of both factors in tumor growth, such as in medulloblastoma, the most frequent malignant brain tumor in children, and in breast cancer has been demonstrated (Di et al., 2005; Terrinoni et al., 2011). Furthermore, OTX1 is involved in the epithelial damage promoting colorectal cancer progression (Yu et al., 2014). Recently, both OTX1 and OTX2 were found to be expressed in normal adult rat small intestine myenteric ganglia: OTX1 has been predominantly found in enteric glial cells and in few myenteric neurons, while OTX2 was uniquely expressed in the soma of a relatively small percentage of myenteric neurons (Filpa et al., 2017a). Expression of both orthodenticle homeobox proteins was upregulated during intestinal ischemia/reperfusion injury and correlated with alterations of the intestinal neuromuscular function in this pathophysiological condition (Filpa et al., 2017a). Such changes involved an interplay between both transcription factors and enteric nitrergic neurons, sustaining the development of altered motor responses involving NO production (Filpa et al., 2017a). In our study, we aim for the first time to evaluate possible changes of OTX1 and OTX2 expression in rat colon myenteric ganglia, after dinitro-benzene sulfonic (DNBS) acid-induced colitis, by means of molecular biology and morphological approaches. In view of the possible adaptive changes induced by the inflammatory insult at remote sites from the injury, we also investigated the consequences of DNBS treatment on both OTX1 and OTX2 expression in small intestine myenteric plexus.

Materials and Methods

Animals

Male Sprague–Dawley rats (weight 250–300 g, Envigo, Udine, Italy), were handled following principles of good laboratory animal care, in accordance with specific national and international laws and regulation. Animals were maintained at a regular 12/12 h light/dark cycle, under controlled environmental conditions (temperature 22 ± 2 °C; relative humidity 60–70%), with free access to a standard laboratory tap water and chow. The protocol was approved by the Animal Care and Use Ethics Committee of the University of Pavia (n. 3/2011).

DNBS-induced colitis

The model of DNBS-induced experimental colitis in rats was chosen since the inflammatory response develops rapidly, reaching the maximum grade in 6 days, and shares many features with the human IBD response (Filpa et al., 2017b). A single dose (30 mg) of 2,4-dinitro-benzene-sulfonic acid (DNBS, ICN Biomedicals, CA, USA), dissolved in 0.25 ml of 50% ethanol, was administered to isofluorane anesthetized rats by means of a polyethylene (PE-60) catheter into the colon 8 cm proximal to the anus. Ethanol, owing to its ability to break the mucosal barrier thus permitting DNBS penetration into the bowel wall, was administered (0.25 ml) to control animals as a vehicle. The dose of DNBS used in the study induces adequate inflammation, without causing unnecessary distress, suffering or mortality to the animals. DNBS-treated and control rats were kept separated during the study. After 6 days, animals were euthanized and the small intestine and distal colon was removed and washed with a physiological Tyrode’s solution (in mM: 137 NaCl; 2.68 KCl; 1.8 CaCl2·2H2O; 2 MgCl2; 0.47 NaH2PO4; 11.9 NaHCO3; 5.6 glucose). Throughout the treatment period, animals were monitored to evaluate possible physiological and behavioral changes (i.e., changes in body weight, respiration, occurrence of diarrhea, alterations of posture and in the appearance of the coat) as marks of suffering and distress.

Assessment of colonic damage

The severity of intestinal inflammation was evaluated macroscopically and histologically. Criteria for the macroscopic evaluation included the presence of adhesions between the colon and other intra-abdominal organs, the extension of hyperemia and macroscopic mucosal damage, the thickening of the colonic wall and the consistency of colonic fecal material, all attributed with a score ranging from 0 (no damage) to 6 (maximum damage) Filpa et al. (2017b).

For the microscopic histological evaluation, standard hematoxylin and eosin (HE) staining was carried out on serial sections (7 μm) of small intestine and colonic segments, from control and DNBS-treated rats, which were first fixed for 24–48 h with 4% formaldehyde in acetate buffer 0.05 M and successively embedded in paraffin. Preparations were observed under a light microscope (Nikon Eclipse Ni; Nikon, Tokyo, Japan) and data were recorded using a DS-5M-L1 digital camera system (Nikon Corporation, Tokyo, Japan).

Myeloperoxidase activity

To evaluate the development of the inflammatory state caused by neutrophil infiltration, myeloperoxidase (MPO) was measured as previously described by Filpa et al. (2017a). Briefly, intestinal segments deprived of the mucosal layer were homogenized (50 mg/mL) with a solution of ice cold potassium phosphate buffer (50 mm, pH 6.0) containing 0.5% hexadecyl trimethylammonium bromide (HTAB). After centrifugation (14,000 rpm, 20 min, 4 °C), an aliquot of the supernatant fraction was mixed with HTAB-phosphate buffer containing O-dianisidine dihydrochloride with hydrogen peroxide before spectrophotometrically recording at 460 nm changes in the rate of absorbance. MPO activity was expressed in units (U)/wet tissue weight, where U defined the amount of enzyme that degrades 1 μmol/min of hydrogen peroxide at 25 °C. Experiments were performed six times for each experimental group.

Immunofluorescence

Paraffin-embedded tissue sections and whole-mount preparations were processed for the immuno-localization of OTX1 and OTX2 on small intestine and colon of control and DNBS-treated rats, as previously described in Bistoletti et al. (2019) and Ceccotti et al. (2018).

Paraffin sections

Seven μm paraffin cross sections incubated with CD45, as a marker of inflammatory infiltration, or either with OTX1 or OTX2 with the pan neuronal marker HUC/D, for double staining. Optimal dilutions of antibodies are reported in Table 1. Coverslips were mounted with Citifluor mounting medium and then observed with fluorescent microscopy (Nikon Instruments, Melville, NY, USA).

Table 1 Primary and secondary antisera and their respective dilutions used for Western Blot (WB) assay and immunohistochemistry (HC).

Antiserum	Dilution (WB)	Dilution (HC)	Source	Host species	
Primary antisera	
OTX1		1:100	Invitrogen (PA5-67901)	Rabbit	
OTX1	1:200	1:100	Santa Cruz (sc133872)	Rabbit	
OTX2	1:100	–	R&D systems (AF1979)	Goat	
OTX2 Alexa Fluor 594 conjugated	–	1:100	Bioss Antibodies (bs-11597R-A594)	Rabbit	
HUC/D biotin	–	1: 100	Invitrogen (16A11)	Mouse	
β-actin	1:1,000	–	Cell Signalling Technology (8H10D10)	Mouse	
nNOS	–	1:200	Millipore (AB1529)	Sheep	
nNOS	–	1:50	Santa Cruz (sc648; R-20)	Rabbit	
iNOS	–	1:50	Santa Cruz (sc8310; H-174)	Rabbit	
S100	–	1:100	Merck Millipore (MAB079-1)	Mouse	
CD45	–	1:100	Merck Millipore (MAB079-1)	Mouse	
Secondary antisera & streptavidin complexes		
Anti-rabbit Alexa Fluor 488	–	1:200	Molecular Probes (A21206)	Donkey	
Anti-mouse Alexa Fluor 488	–	1:200	Molecular Probes (A21202)	Donkey	
Cy3-conjugated streptavidin	–	1:500	Amersham (PA43001)		
FITC-conjugated straptavidin	–	1:200	Molecular Probes (SA1001)		
Anti-rabbit IgG HRP peroxidase conjugated	1:5,000	–	Santa Cruz (sc2004)	Goat	
Anti-goat IgG HRP peroxidase conjugated	1:10,000	–	Santa Cruz (sc2020)	Donkey	
Anti-mouse IgG, HRP-linked	1:1,000	–	Cell Signalling Technology (#7076)	Horse	
Note:

Supply companies: Amersham, GE Healthcare, Buckinghamshire, UK; Bioss antibodies, MA, USA; Cell Signaling Technology, Danvers, MA, USA; Invitrogen, Thermo Fisher, MA, USA; Merck Millipore Burlington, MA, USA; Molecular Probes, Thermo Fisher, MA, USA; R&D systems, Minneapolis, CA, USA; Santa Cruz Biotechnology, CA, USA; Sigma–Aldrich, Milano, Italy.

Whole-mount preparations

Whole-mount preparation immunolabeling was performed on the longitudinal muscle with the attached myenteric plexus (LMMP) proceeding with a double-staining with optimally diluted primary and secondary antibodies (Table 1). Preparations were mounted onto glass slides with a mounting medium (Vectashield with DAPI; Vector Lab, Burlingame, CA, USA) and analyzed with Image J NIH image software (http://imagej.nih.gov/ij). Total neuron number per ganglion area was expressed as the ratio between the number of neurons positive for HuC/D and the total ganglion area (µm2). Areas of 15–19 myenteric ganglia from LMMP preparations of 5 vehicle-treated CTR and 5 DNBS-treated animals were measured at 40× magnification by tracing boundaries around stained cell somas (HuC/D) (Filpa et al., 2017a). Neuronal cell body area was measured with Image J. The number of OTX1 or OTX2 immunoreactive neurons that co-localized with HuC/D were counted from a total of 10–15 ganglia (5 animals for each experimental group). The proportion of OTX1 and OTX2 immunoreactive myenteric neurons, was expressed as percentage of the total number of HuC/D positive neurons (Filpa et al., 2017a). To evaluate negative controls and interference control staining, both primary and secondary antibodies were omitted, incubating the colonic whole-mounts with non-immune serum from the same species in which the primary antibodies were raised. In all these conditions, no specific signal was detected. Pictures from preparations, collected with a Leica TCS SP5 confocal laser scanning system (Leica Microsystems GmbH, Wetzlar, Germany) were then processed with Adobe-Photoshop CS6S software.

Real time quantitative RT-PCR

To evaluate the influence of DNBS-induced colitis on OTX1, OTX2, TNFα, pro-IL1β, IL6, HIF1α, VEGFα, mRNA levels, total RNA from rat small intestine and colon LMMPs was extracted with TRIzol (Invitrogen, Carlsbad, CA, USA) and treated with DNase I (DNase Free, Ambion) to remove possible traces of contaminating DNA. 2.5 μg of total RNA was retrotranscribed using the High Capacity cDNA synthesis kit (Applied Biosystems, Life Technologies, Grand Island, NY, USA). Quantitative RT-PCR was performed on the Abi Prism 7000 real-time thermocycler (Applied Biosystems, Foster City, CA, USA) with Power Sybr Green Universal PCR Master Mix (Applied Biosystems, Foster City, CA, USA) following manufacturer’s instructions. Primers were designed to have a similar amplicon size and similar amplification efficiency as required for the utilization of the 2−ΔΔCt method to compare gene expression (Bin et al., 2018), using Primer Express software (Applied Biosystems, Foster City, CA, USA) on the basis of available sequences deposited in public database (Table 2). For quantitative RT-PCR a final concentration of 500 nm for each primer was used. Experiments were performed at least 7 times for each experimental group. 2−ΔΔCt values obtained from the comparison between normalized Ct values of DNBS-treated samples with those obtained from control samples were used to evaluate the effect of DNBS-induced colitis on OTX1 and OTX2 mRNA levels in the small intestine and colon.

Table 2 Sequence of primers used in the study for the analysis of qRT-PCR.

Gene	Sequence	
β-actin	F 5′-AGGCCCCTCTGAACC-3′	
R 5′-GGGGTGTTGAAGGTC-3′	
OTX1	F 5′-GCGAGGAGGTGGCTCTCA-3′	
R 5′-GGCTCGGCGGTTCTTGA-3′	
OTX2	F 5′-CCCAATTTGGGCCGACTT-3′	
R 5′-GCGTAAGGCGGTTGCTTTAG-3′	
TNFα	F 5′-CCCAGACCCTCACACTCAGAT-3′	
R 5′-TTGTCCCTTGAAGAGAACCTG-3′	
pro-IL1β	F 5′-CCCTGCAGCTGGAGAGTGTGG-3′	
R 5′-TGTGCTCTGCTTGAGAGGTGCT-3′	
IL6	F 5′-GTGCAATGGCAATTCTGATTGTA-3′	
R 5′-CTAGGGTTTCAGTATTGCTCTGA-3′	
VEGFα	F 5′-GCTGTGTGTGTGAGTGGCTTA-3′	
R 5′-CCCATTGCTCTGTACCTTGG-3′	
HIF1α	F 5′-AAGCACTAGACAAAGCTCACCTG-3′	
R 5′-TTGACCATATCGCTGTCCAC-3′	

Western immunoblot analysis

Purified membrane fractions obtained after successive centrifugations of homogenized intestinal LMMPs preparations were used to analyze OTX1 and OTX2 protein level as described elsewhere (Giaroni et al., 2011). Membrane incubation with OTX1 and OTX2 primary antibodies was performed by incubation with a horseradish peroxidase-conjugated secondary antisera (Table 1). The signal of antibody/substrate complex was visualized by chemiluminescence using an enhanced chemiluminescence kit (ECL advance Amersham Pharmacia Biotech, Cologno Monzese, Italy), and then evaluated by densitometric analysis using Image J NIH image software. The effect of DNBS-induced colitis on OTX1 and OTX2 protein levels was expressed as the percentage variation of the optical density (expressed in arbitrary units) of OTX1 and OTX2 signals normalized to the respective β-actin, used as protein loading control, in LMMP preparations obtained from DNBS-treated animals compared to controls. Experiments were performed at least five times for each experimental group. Specificity of OTX1 and OTX2 primary antibodies was evaluated by testing their selectivity in spontaneous immortalized human Müller cell line (MIO-M1) and in rat hippocampus, respectively (data not shown) (Filpa et al., 2017a). Negative controls were performed by omitting the primary antibody.

Statistical analysis

All data are expressed as mean ± S.E.M. Statistical significance was calculated by either Student’s t test or by one-way ANOVA with Tukey’s post hoc test, where appropriate, using GraphPad Prism (version 5.3; GraphPad Software, San Diego, CA, USA). Differences among groups were considered significant when P value were 0.05 or lower.

Results

Assessment of colitis

Macroscopic assessment

Body weight was significantly reduced in DNBS-treated rats compared to non-inflamed controls at day 6 after treatment (DNBS-treated: 280 ± 6.50 g, n = 7; vehicle-treated CTR: 306 ± 2.3 g, n = 7, P = 0.027 by Student’s t test). Six days after DNBS administration, the distal colon was thickened and ulcerated with evident regions of transmural inflammation, adhesions between the colon and other intra-abdominal organs were often present and the bowel was occasionally dilated. Macroscopic damage score significantly increased in the colon of DNBS-treated animals in comparison with vehicle-treated controls (DNBS-treated: 4.80 ± 021, n = 10; CTR: 0.3 ± 0.08, n = 14 P < 0.0001, by Student’s t test). No significant gross morphology changes were observed in the small intestine. This was reflected by low macroscopic damage scores in both experimental groups (DNBS-treated: 0.11 ± 0.08, n = 10; CTR: 0.09 ± 0.05, n = 14, by Student’s t test). These findings are in agreement with other reports, showing that after induction of an experimentally-induced colitis, the occurrence of gross macroscopic changes is obvious only on the site of injury, preserving distant sites along the gastrointestinal tract (Mourad et al., 2016).

Histological assessment

A further set of experiments was carried out on intestinal cross sections to evaluate the impact of DNBS–induced colitis on the architecture of the rat small intestine and distal colon by means of standard HE staining, and the degree of inflammatory infiltrate by CD45 immunofluorescent staining. Small intestine and distal colon cross-sections obtained from control animals showed normal histological features, with compact myenteric ganglia, formed by healthy neuron and glial cells, laying between the circular and longitudinal muscle of the muscularis propria (Figs. 1A, 1B, 1K and 1L). CD45 staining, indicative of inflammatory cell infiltration, was almost negligible in both small intestine and colon control cross sections (Figs. 1C and 1M). At day 6 after DNBS treatment, small intestine specimens did not show prominent histological abnormalities (Figs. 1F and 1G). In contrast, both mucosa and serosal epithelium of the distal colon obtained from DNBS-treated rats, displayed morphological abnormalities (Figs. 1P–1R). The colonic mucosal surface was irregular and crypt architecture was profoundly altered, the muscularis mucosae, submucosa and muscularis propria layers were thickened (Fig. 1P). Prominent spaces between smooth muscle cells and leukocyte infiltration were also observed (Fig. 1Q). In the small intestine of DNBS-treated animals myenteric neurons had a normal morphology comparable to that observed in control preparations as shown Fig. 1G. In contrast, colonic myenteric ganglia underwent important degenerative changes, with neurons displaying cytoplasm vacuolization and irregular nuclear and cellular membrane. Large spaces between muscle cells were also evident (Fig. 1R). After DNBS-treatment, CD45 staining significantly increased both in the small intestine and in the colon (Figs. 1H and 1S).

Figure 1 Histological evaluation, OTX1 and OTX2 staining in small intestine and colonic cross sections after DNBS-induced colitis.

Hematoxylin-eosin (HE) staining highlights the well-preserved cellular and structural morphology of small intestine (A) and distal colon (K) cross-sections of control (vehicle-treated, CTR) rats (bar: 50 µm). Detail of a myenteric ganglion in cross-sections obtained from small intestine (B) and colon (L) of CTR rats (bars: 10 µm). Both in the small intestine (C) and distal colon (M) of CTR animals, CD45 staining, as an index of inflammatory infiltrate, was slight (bars: 10 µm). After DNBS treatment no significant morphological changes were evident in the small intestine (F), while important structural changes were observed in the distal colon (P), including increased thickness of the muscularis mucosae, submucosal layer and muscularis propria layers (bars: 50 µm). Detail of a myenteric plexus ganglion in small intestine (G) and distal colon (R) cross-sections obtained after DNBS treatment (bars: 10 µm). In particular, HE staining highlights prominent distortions in a colonic myenteric plexus ganglion (R). (Q) Shows the presence of spaces among smooth muscle cells and leukocytes infiltrates (arrowheads) in a colonic cross-section after DNBS treatment (bar 10 µm). After DNBS treatment, CD45 significantly increased both in the small intestine (H) and in the colon (S). Co-staining with the pan neuronal marker HuC/D, showed few myenteric neurons labeling for both OTX1 and OTX2 (arrows) in the small intestine (D and N) and colon (E and O) of CTR rats, inserts show details of myenteric ganglia (bars: 10 µm; inserts 5 µm). After DNBS treatment, both OTX1 and OTX2 immunoreactivity increased in myenteric neurons of both regions (small intestine: (I and T); distal colon: (J and U); bars: 10 µm, inserts 5 µm). Arrows indicate myenteric plexus ganglia. CM, circular muscle; LM, longitudinal muscle; MP, myenteric plexus; M, mucosa; SM, submucosa; Mus, Muscularis Propria.

We then focused our histological investigations on myenteric ganglia by carrying out immunofluorescence staining of small intestine and distal colon LMMP whole-mount preparations with the pan neuronal marker HuC/D. In this set of experiments changes in myenteric neuron number were observed in both gastrointestinal regions after DNBS treatment (Figs. 2A–2F). In particular, the number of myenteric neurons staining for HuC/D was significantly reduced (P < 0.05, by one way-ANOVA with Tukey’s post hoc test) with respect to values obtained in respective control preparations in both regions (Fig. 2E). Myenteric neuron number was reduced to a significantly greater extent (P < 0.05, by one way-ANOVA with Tukey’s post hoc test) in the colon than in the small intestine (Fig. 2E). In addition, in the colon, but not in the small intestine, DNBS-induced inflammation was associated with a significant reduction of myenteric neuron soma area (P < 0.001, by one way-ANOVA with Tukey’s post hoc test) compared to control preparations (Fig. 2F). As expected, in accordance with previous studies, in control preparations, the number of myenteric neurons was significantly lower in the colon (P < 0.05, by one way-ANOVA with Tukey’s post hoc test) than in the small intestine (Fig. 2E). Additionally, in the distal colon the neuronal soma area was significantly larger (P < 0.05, by one way-ANOVA with Tukey’s post hoc test) than in the small intestine (Fig. 2F). Overall, these observations suggest that after DNBS-induced colitis, major histological changes involving myenteric ganglia, occur in the distal colon. However, also the small intestine may be influenced by the injury, as indicated by the increased infiltration of inflammatory CD45+ cells and by the reduction of myenteric neuron number.

Figure 2 DNBS treatment-induced changes in myenteric neurons.

(A–D) HuC/D staining of myenteric neurons in CTR and DNBS-treated whole-mount LMMP preparations. (E) Myenteric neuron number normalized per ganglion area in small intestine and distal colon LMMP whole-mount preparations obtained from DNBS-treated (solid bar) and control animals (vehicle-treated CTR, empty bar). (F) Mean myenteric neuron area calculated in small intestine and distal colon whole-mount preparations obtained from DNBS-treated (solid bar) and control animals (vehicle-treated CTR, empty bar). Values are expressed as mean ± S.E.M. N = 5 rats per group. *P < 0.05 and ***P < 0.001 vs. values obtained in CTR animals by One Way ANOVA with Tukey’s post hoc test. °P < 0.05 vs. CTR small intestine. §P < 0.05 vs. DNBS-treated small intestine.

Biomolecular assessment

The occurrence of an inflammatory response in LMMP preparations, on site and distantly from the injury, was further investigated by evaluating the levels of the mRNA of different cytokines, which are considered standard markers of intestinal inflammation, and by measuring MPO activity. In both intestinal regions, after DNBS treatment, a significant enhancement of the expression of inflammatory cytokines TNFα (P < 0.001 in both regions, by one-way ANOVA with Tukey’s post hoc test), pro-IL1β (P < 0.01 in the ileum and P < 0.001 in the colon, by one way-ANOVA with Tukey’s post hoc test), IL6 (P < 0.001 in both regions by one-way ANOVA with Tukey’s post hoc test), and of HIF-1α (P < 0.05 in both regions, by one-way ANOVA with Tukey’s post hoc test) and VEGFα (P < 0.01 in both regions, by one-way ANOVA with Tukey’s post hoc test), was evidenced with respect to control animals (Figs. 3A–3E). As regards the pro-inflammatory cytokines, enhancement of TNFα mRNA was similar in the colon and in the small intestine, while pro-IL1β and IL-6 mRNA levels were significantly higher in the colon with respect to the small intestine (P < 0.05 and P < 0.001, respectively, by one-way ANOVA with Tukey’s post hoc test) (Figs. 1A–1C). HIF1α and VEGFα mRNA levels after DNBS treatment was not significantly different in the small intestine with respect to the distal colon (Figs. 1D and 1E).

Figure 3 Expression of Inflammatory markers in the ileum and colon after DNBS-induced colitis.

(A–E) qRT-PCR quantification of TNFα, pro-IL1β, IL6, HIF1α, VEGFα mRNA levels obtained in the small intestine and distal colon of control animals (vehicle-treated, CTR) and after DNBS treatment (solid bars). The relative gene expression was determined by comparing 2−ΔΔCt values in CTR and DNBS-treated samples normalized to β-actin. Values are mean ± S.E.M. of at least 7–9 experiments. *P < 0.05, **P < 0.01 and ***p < 0.0001 vs. values obtained in control animals; §P < 0.05 and §§§P < 0.001 vs. DNBS-treated small intestine. Significance was evaluated by one-way ANOVA with Tukey’s post hoc test. (F) MPO activity measured in mucosa-deprived small intestine and distal colon segments obtained from DNBS-treated (solid bar) and control animals (vehicle-treated, CTR). Values are expressed as mean ± S.E.M. of 6 experiments. ***P < 0.001 vs. values obtained in control animals, by one-way ANOVA with Tukey’s post hoc test.

MPO activity significantly increased in rat small intestine (P < 0.001, by one way-ANOVA with Tukey’s post hoc test) and colonic (P < 0.001, by one way-ANOVA with Tukey’s post hoc test) segments compared to control animals, suggesting the occurrence of inflammation-induced neutrophil infiltration on site and distally from the injury (Fig. 1F).

Distribution of OTX1 immunoreactivity in small intestine and colon LMMP whole-mount preparations from normal and DNBS-treated animals

In view of the adaptive changes observed after DNBS-induced inflammation in the rat myenteric plexus, we subsequently evaluated if this adaptation encompasses alterations in the distribution of homeobox proteins involved neuronal plasticity, OTX1 and OTX2, in cross sections and LMMP whole-mounts of the small intestine and distal colon. In cross-sections of small intestine and distal colon obtained from vehicle-treated control rats, myenteric ganglia displayed faint OTX1 immunoreactivity (Figs. 1D and 1N). In control LMMP whole-mount preparations OTX1 antibody stained few myenteric neurons in both the small intestine (3.79 ± 1.00%, n = 15) and distal colon (4.77 ± 1.43%, n = 15) (Fig. 4A). In both intestinal regions, OTX1 antibody stained the soma and nucleus of large and medium size myenteric neurons with either a round or an ovoidal shape (Figs. 5A–5C and 5G–5I). In small intestine and distal colon cross-sections obtained from DNBS-treated animals, a significant increase of OTX1 staining was observed within the myenteric plexus with respect to control preparations (Figs. 1I and 1T). After DNBS treatment, a significant increase in the number of OTX1 immunoreactive myenteric neurons was observed in whole-mount preparations of both small intestine (9.86 ± 0.86%, n = 15, P < 0.0001, by one-way ANOVA with Tukey’s post hoc test) and colon (40.25 ± 1.2%, n = 15 P < 0.0001, by one-way ANOVA with Tukey’s post hoc test) with respect to control preparations (Figs. 4A, 5D–5F and 5J–5L). In DNBS-treated colonic whole mount preparations, the percentage of OTX1 immunoreactive myenteric neurons was significantly higher than in the small intestine (Fig. 4A). In LMMP whole-mount preparations of both intestinal regions, OTX1 antibody co-localized with the glial marker S100β, and co-staining was particularly intense after DNBS-induced inflammation (Figs. 5M–5R). In small intestine and colonic LMMPs obtained from control and DNBS-treated animals, myenteric neurons staining for OTX1 were also immunopositive for iNOS, as shown in Figs. 6A–6L. Both in control and DNBS-treated LMMP whole-mount preparations, OTX1 immunostaining did not co-localize with nNOS (Figs. 6M–6O). These results suggest that OTX1 is expressed both in myenteric neurons, where it is highly co-expressed with iNOS, and in enteric glial cells of the rat small intestine and distal colon. After DNBS-induced colitis the number of myenteric neurons expressing OTX1 is highly upregulated both on site and distantly.

Figure 4 Percentage of OTX1 and OTX2 neurons in the rat small intestine and distal colon after DNBS-treatment.

(A) Percentage of OTX1 immunoreactive myenteric neurons with respect to the total of HuC/D immunoreactive neurons per ganglion in small intestine and distal colon LMMPs whole-mount preparations obtained from DNBS-treated (solid bar) and control animals (vehicle-treated CTR, empty bar). (B) Percentage of OTX2 immunoreactive myenteric neurons with respect to the total of HuC/D immunoreactive neurons per ganglion in small intestine and distal colon whole-mount preparations obtained from DNBS-treated (solid bar) and control animals (vehicle-treated CTR, empty bar). Values are expressed as mean ± S.E.M, N = 5 rat per group. ***P < 0.001 vs. values obtained in CTR animals, §§§P < 0.001 vs. DNBS-treated small intestine by one-way ANOVA with Tukey’s post hoc test.

Figure 5 OTX1 localization in the rat small intestine and distal colon myenteric plexus of CTR and DNBS-treated rats.

Co-localization of OTX1 with the neuronal marker HuC/D in LMMPs whole-mount preparations of the small intestine and distal colon obtained from vehicle-treated CTR animals (A–C; G–L, respectively) and DNBS-treated animals (D–F; J–L, respectively). In CTR preparations few neurons displayed OTX1 immunoreactivity (arrow) (A–C; G–I). In both regions, the number of OTX1-IR neurons was higher in DNBS-treated preparations (D–F; J–L). Photomicrographs of small intestine (M–O) and colonic (P–R) LMMP whole-mount preparations showing OTX1 immunoreactivity in enteric glial cells (asterisk) as evidenced by co-staining with the glial cell marker, S100β. Bar 50 µm.

Figure 6 OTX1 and iNOS co-localization in the rat small intestine and distal colon myenteric plexus of CTR and DNBS-treated rats.

Photomicrograph showing few myenteric neurons (arrow) co-staining OTX1 and iNOS in CTR LMMPs whole-mount preparations of the small intestine (A–C) and distal colon (G–I). After DNBS treatment, the number of iNOS immunoreactive myenteric neurons (arrow) increased and co-expressed OTX1 both in small intestine (D–F) and distal colon (J–L). Both in CTR and DNBS-treated group, co-staining between OTX1 and nNOS was not detected in myenteric ganglia of both small intestine and distal colon. A colonic whole-mount LMMP preparation obtained from DNBS-treated animals is reported as an example (M–O, bar: 50 µm) bar: 50 µm.

Distribution of OTX2 immunoreactivity in small intestine and colon LMMP whole-mount preparations from normal and DNBS-treated animals

In control sections of the rat small intestine and colon, faint OTX2 staining was found in myenteric ganglia (Figs. 1E and 1O). In LMMPs whole-mount preparations obtained from control animals OTX2 specific antibody stained the soma of few myenteric neurons both in the small intestine and in the distal colon (Figs. 7A–7C and 7G–7I, respectively). Some neurons had a large cell body, while other smaller neurons with an ovoidal shape were, prevalently, localized in the periphery of the ganglion. The percentage of OTX2-immunoreactive neurons in control small intestine and distal colon myenteric plexus was 7.07 ± 1.68%, n = 15 and 8.93 ± 2.11%, n = 15, respectively (Fig. 4B). In small intestine and colon cross-sections obtained from DNBS-treated animals, OTX2 staining significantly increased in myenteric ganglia with respect to control preparations (Figs. 1J and 1U). Accordingly, after DNBS treatment, the percentage of OTX2 immunoreactive myenteric neurons in LMMP whole-mount preparations obtained from the small intestine and colon significantly increased (47.95 ± 3.20%, n = 15; 41.47 ± 5.12%, n = 15, respectively P < 0.001, by one-way ANOVA with Tukey’s post hoc test) with respect to control values (Figs. 4B, 7D–7F and 7J–7L). The number of OTX2 immunoreactive neurons was not significantly different in the two regions (Fig. 4B). In small intestine and colonic myenteric ganglion obtained from control and DNBS-treated animals, the majority of OTX2 immunoreactive neurons stained for nNOS (Figs. 8A–8L). There was no evidence of co-staining between OTX2 and iNOS, as shown in Figs. 8M–8O. In small intestine and distal colon LMMPs obtained from control animals 31.94 ± 3.33%, n = 10 and 30.83 ± 10.83%, n = 10, respectively, of OTX2-IR myenteric neurons stained also for OTX1. In both regions, after DNBS treatment, 25.82 ± 5.78%, n = 10 and 25.63 ± 2.17% n = 10, respectively, of OTX2-IR were also immunoreactive for OTX1. These findings suggest that OTX2 is exclusively expressed in myenteric neurons and displays a high co-localization with nNOS. Experimentally-induced colitis upregulates the number of myenteric neurons expressing OTX2 both on site and distantly.

Figure 7 OTX2 localization in the rat small intestine and distal colon myenteric plexus of CTR and DNBS-treated rats.

Co-localization of OTX2 with the neuronal marker HuC/D in LMMPs whole-mount preparations of the small intestine and distal colon obtained from vehicle-treated CTR (A–C; G–I, respectively) and DNBS-treated animals (D–F; J–L, respectively). In CTR preparations, few neurons displayed OTX2 immunoreactivity (arrow). In both regions, after DNBS treatment, the number of OTX2 immunoreactive neurons significantly increased. Bar 50 µm.

Figure 8 OTX2 and nNOS co-localization in the rat small intestine and distal colon myenteric plexus of CTR and DNBS-treated rats.

In myenteric neurons of both small intestine and distal colon from CTR and DNBS-treated rats, OTX2 co-localized with nNOS (arrows). Some OTX2 immunoreactive neurons, however, did not stain for nNOS (asterisk) (A–L, bars: 50 μm). Co-staining between OTX2 and iNOS was not detected, in myenteric ganglia of both small intestine and distal colon of both experimental groups. A colonic whole-mount LMMP preparation obtained from DNBS-treated animals is reported as an example (M–O, bar: 50 µm).

DNBS-induced colitis upregulates OTX1 mRNA and protein in LMMPs of the rat small intestine and distal colon

In order to confirm the immunofluorescence data reflecting an up-regulation of OTX1 and OTX2 in the rat small intestine and colon myenteric plexus after DNBS-induced colitis, we evaluated the levels of mRNA and proteins of both homeobox proteins in LMMP preparations. Rat small intestine and colonic OTX1 mRNA levels obtained in the different experimental groups are described in Fig. 9A. In LMMPs of both regions, DNBS treatment induced a significant increase of OTX1 mRNA with respect to values obtained in control preparations (small intestine: P < 0.05; distal colon: P < 0.001, by one-way ANOVA with Tukey’s post hoc test). OTX1 mRNA enhancement in colonic specimens was significantly higher than in the small intestine (P < 0.001, by one-way ANOVA with Tukey’s post hoc test). In rat small intestine and colonic LMMP preparations the specific OTX1 antibody revealed one band at 37 kDa (Fig. 9B). In both gut regions, after DNBS treatment, OTX1 protein expression significantly increased (small intestine: P < 0.05; distal colon: P < 0.01, by one-way ANOVA with Tukey’s post hoc test). These biomolecular data confirm that an experimentally-induced colitis may up-regulate both the transcript and protein of OTX1 in the myenteric plexus both on site and distantly from the injury.

Figure 9 qRT-PCR and western blot analysis of OTX1 and OTX2 mRNA and protein levels in the rat small intestine and colon after DNBS-induced colitis.

RT-PCR quantification of OTX1 (A) and OTX2 (C) transcripts in preparations of the small intestine and distal colon obtained from control (CTR, empty bars) and DNBS-treated animals (solid bars). The relative gene expression was determined by comparing 2−ΔΔCt values in CTR and DNBS-treated samples normalized to β-actin. Values are mean ± S.E.M. of 7–13 experiments. *P < 0.05, ***P < 0.001 vs. CTR animals; §§§P < 0.001 vs. DNBS-treated small intestine by one-way ANOVA with Tukey’s post hoc test. (B) OTX1 and (D) OTX2 protein expression analyzed in LMMPs preparations of the small intestine and colon obtained from CTR (empty bars) and DNBS-treated animals (solid bars). Blots representative of immunoreactive bands for either OTX1 or and β-actin in the different experimental conditions are reported on top of each panel. Numbers at the margins of the blots indicate relative molecular weights of the respective protein in kDa. Samples (200 μg) were electrophoresed in SDS-8% polyacrylamide gels. Values are expressed as mean ± S.E.M. of 5–6 experiments of the percentage variation of the normalized optical density (O.D.) obtained from DNBS-treated preparations with respect to values obtained in control samples. *P < 0.05 ad **P < 0.01 vs. CTR by one-way ANOVA with Tukey’s post hoc test.

DNBS-induced colitis upregulates OTX2 mRNA and protein in LMMPs of the rat small intestine and distal colon

OTX2 mRNA levels measured in rat small intestine and colonic LMMPs preparations obtained from control and DNBS-treated animals are described in Fig. 9C. In both regions, DNBS treatment induced a significant increase of OTX2 mRNA with respect to values obtained in control preparations (small intestine: P < 0.05; distal colon: P < 0.001, by one-way ANOVA with Tukey’s post hoc test). OTX2 mRNA levels in small intestine specimens were not significantly different than those obtained in the distal colon. In rat small intestine and colon LMMP preparations the specific OTX2 antibody revealed one band at 32 kDa (Fig. 9D). In both gut regions, OTX2 protein expression significantly increased (P < 0.01, by one-way ANOVA with Tukey’s post hoc test) after DNBS treatment. After DNBS-induced colitis, analogously to OTX1, up-regulation of OTX2 in the rat myenteric plexus was evidenced also for the transcript and protein, by means of qRT-PCR and western immunoblotting.

Discussion

In myenteric neurons, intestinal inflammation induces adaptive changes, which may be responsible for both acute and long-lasting alterations of the gastrointestinal functions, underlying disease symptoms (Brierley & Linden, 2014). The molecular mechanism/s of enteric neuronal adaptation to inflammation are still largely to be uncovered. In this study, we show that the expression of orthodenticle homeoproteins, OTX1 and OTX2, which participate to neuroplastic changes both in physiological and pathological conditions (Spatazza et al., 2013), is up-regulated in the myenteric plexus of the rat small intestine and distal colon after an experimentally induced colitis with DNBS acid.

In analogy with the results recently obtained in the rat small intestine myenteric plexus (Filpa et al., 2017a), in the present study OTX1-immunoreactivity (IR) is predominantly found in enteric glial cells and in few neurons of the rat distal colon myenteric plexus, while OTX2-IR is uniquely detected in the soma of a relatively small percentage of myenteric neurons. In pathophysiological conditions, that is, after in vivo-induced ischemia/reperfusion (I/R) injury as well as in mild inflammatory conditions induced in sham-operated animals OTX1 and OTX2 expression in rat small intestine myenteric ganglia significantly enhanced (Filpa et al., 2017a). In good agreement, in the present study we show a significant increase of both OTX1 and OTX2 mRNA and protein levels in LMMP preparations of the rat small intestine and distal colon myenteric plexus after DNBS-induced colitis. In addition, the number of myenteric neurons staining for both transcription factors in the two intestinal regions significantly increases, suggesting that the inflammation influences the expression of both homeoproteins in myenteric neurons, on site and distantly from the injury. This adaptive response is not associated with major histopathological alterations in the small intestine myenteric plexus. In line with previous data, DNBS treatment, induces important morphological and histological changes, including distortion of myenteric ganglia with signs of neuronal degeneration, in the colon, but not in the small intestine, where, however, the number of myenteric neurons significantly decreases. Interestingly, MPO activity and the number of CD45 positive cells, which are indicative of inflammatory infiltration, are elevated both in small intestine and distal colon segments. In addition, after DNBS-induced colitis, in rat colonic and small intestine LMMPs the levels of pro-inflammatory cytokines, such as TNFα, pro-IL1β and IL6, and of factors involved in active intestinal inflammation, such as HIF1α and VEGFα, are up-regulated, as observed along the rat gastrointestinal tract after trinitrobenzene sulfonic (TNBS) acid- and DNBS-induced colitis by other groups (Bakirtzi et al., 2014; Barada et al., 2006). Overall, our observations suggest that important molecular changes, including upregulation of OTX1 and OTX2 pathways, may occur in the myenteric plexus along the gastrointestinal tract in response to an inflammatory challenge. Such changes may be transmitted from the site of injury to distant sites, where overt structural changes are not evident, and may account for the functional alterations observed in the proximal intestine in both IBD patients and animal models, including altered motility (Brierley & Linden, 2014; De Schepper et al., 2008; Blandizzi et al., 2003). We cannot exclude that inflammation-induced OTX1 and OTX2 up-regulation in rat small intestine and distal colon myenteric neurons may influence the neuromuscular function by regulating enteric neurotransmitter pathways. Indeed, TNBS-induced colitis in the rat was associated with a decrease of neurotransmitter release, not only in the inflamed colon, but also distally, in the small intestine resulting in the reduction of small intestinal transit (Blandizzi et al., 2003). Nitric oxide (NO), may represent a putative enteric neurotransmitter involved in OTX1 and OTX2 inflammation-induced upregulation. Noteworthy, a preferential involvement of enteric nitrergic pathways underlays development of dysmotility after deletion of a homeobox gene phylogenetically related to OTX, such as Ncx/Hox11L.1 (Kobayashi et al., 2007). In addition, the occurrence of an interplay between enteric nitrergic pathways and OTX transcription factors has been suggested to sustain nitric oxide (NO)-mediated dysmotility after I/R injury in the gut (Filpa et al., 2017a). In these conditions, NO was shown to favor OTX1 and OTX2 up-regulation in rat small intestine myenteric ganglia. After I/R, NO derived from iNOS promoted OTX1 up-regulation more, while nNOS more closely related to OTX2 up-regulation. During gut inflammation and I/R injury, iNOS and nNOS have different roles on enteric neuronal homeostasis, retaining a neurodamaging and neuroprotective role, respectively (Rivera et al., 2012; Filpa et al., 2017a; Bódi, Szalai & Bagyánszki, 2019). Activation of the inducible isoform and downregulation of nNOS, are correlated with derangement of the neuromuscular function and slowing of the gastrointestinal transit (Rivera et al., 2012; Giaroni et al., 2013; Filpa et al., 2015). In view of the relationship between the two NO synthases and OTX1 and OTX2 these effects on intestinal motility may involve corresponding activation of molecular pathways downstream of OTX1 and OTX2, respectively (Filpa et al., 2017a). Accordingly, in the present study, a substantial involvement of iNOS and nNOS in colitis-induced upregulation of OTX1 and OTX2, respectively, is suggested by the superimposition of OTX1 with iNOS and OTX2 with nNOS immunostaining, in myenteric ganglia of both regions studied. It is possible that activation of the iNOS-OTX1 pathway both in neurons and glial cells may sustain inflammation-induced alterations of the enteric neuromuscular function, as observed after I/R injury (Filpa et al., 2017a). Interestingly, overexpression of OTX1 sustained human colorectal cancer cell proliferation and invasion in vitro and tumor growth in vivo, suggesting a pathogenic role for this homeoprotein in colon cancer (Yu et al., 2014). By contrast, up-regulation of OTX2 in nNOS+ myenteric neurons may have a protective role, preventing inflammation-induced neuronal cell death, as suggested for OTX2 expression in photoreceptors (Housset et al., 2013). Different hypotheses may be put forward to explain the mechanism/s underlying NO-mediated up-regulation of OTX1 and OTX2. This latter may depend on increased levels of NO-derived reactive oxygen species (ROS), whose production in myenteric neurons enhances as a consequence of several gut pathophysiological conditions (Carpanese et al., 2014; Thrasivoulou et al., 2006). In this context, OTX2 up-regulation was recently observed in bovine in vitro embryos as a result of increased oxidative stress inducing ROS production (Leite et al., 2017). We cannot, however, exclude that OTX1 and OTX2 up-regulation in our model may depend upon neuroimmune interactions between different subpopulation of enteric neurons and immunocytes (De Schepper et al., 2008). Neuronal cells in the ENS are located in close proximity to mucosal immunocytes and may regulate one another’s functions by releasing a complex set of cytokines, neurotransmitters and hormones (Lakhan & Kirchgessner, 2010). OTX1 and OTX2 expression in the colonic myenteric plexus may be influenced by inflammatory mediators, such as VEGFα, which is positively correlated to OTX2 expression in retinal pigment epithelial cells during inflammation (Azzolini et al., 2013). More recently, TNFα has been proposed as a modulator of OTX2 expression in in vitro models of chronic subretinal inflammation (Mathis et al., 2017). In addition, a recent genomewide study has identified a positive correlation between inflammatory cytokines, particularly IL6, and OTX1, in the pathogenesis of foot-and-mouth viral disease in animals (Zhang et al., 2018). Circulating cytokines and inflammatory mediators may influence myenteric neurons even at sites distant from the lesion, thus sustaining OTX1 and OTX2 up-regulation in the small intestine myenteric plexus (Liu et al., 2000). Changes in myenteric neuron responses during inflammation may be conveyed also via neuronal pathways constituting local or extrinsic reflex arches, as demonstrated in gastroparesis associated with TNBS-induced ileitis in rats (Moreels et al., 2001). OTX proteins may be secreted and internalized by live cells, including neurons, and a retrograde transfer of these proteins from colonic myenteric neurons to those of the small intestine by peculiar nonconventional mechanisms is also feasible (Joshi et al., 2011).

The pathophysiology of intestinal inflammation reflects a balance between mucosal injury and repair mechanism and may have important consequences on the integrity of intrinsic neuronal circuitries constituting the ENS. Several studies are focused on molecular pathways underpinning damage in different cellular populations, including myenteric neurons. Development of new therapeutic strategies in this field comprise the modulation of inflammation-induced cell death pathways (Baj et al., 2019a, 2019b). In this view our data, by providing new hints on possible regulatory genes involved in myenteric neurons response to an inflammatory injury may contribute to the development of novel strategies for the treatment of gastrointestinal diseases with important social and clinical impact, such as IBD.

Conclusions

In this study we provide evidence that DNBS treatment in rats may elevate inflammatory markers not only in the site of inflammation, but also distally, in the small intestine. Such response is associated with changes in myenteric neuron number in both regions, although more severe damage occurs in the distal colon. After DNBS treatment the expression of two homeobox transcription factors, OTX1 and OTX2, is upregulated in the myenteric plexus of the small intestine and distal colon.

Supplemental Information

Supplemental Information 1 Raw data.

Click here for additional data file.

Supplemental Information 2 Full-length gel of OTX2 and b-actin.

Click here for additional data file.

Supplemental Information 3 Full-length gel of beta actin for OTX1.

Click here for additional data file.

Supplemental Information 4 Full-length gel of OTX1.

Click here for additional data file.

The Authors wish to thank Mr. Antonio Pelizzoli for the technical support. Dr. Fabrizio Bolognese, Dr. Ivan Vaghi, are kindly acknowledged for the excellent assistance in the acquisition of confocal images. Michela Bistoletti, Giovanni Micheloni and Annalisa Bosi are PhD students of the “Experimental and Translational Medicine” course at the University of Insubria. Nicolò Baranzini is a PhD student of the Biotechnology, Biosciences and Surgical Technology” course at the University of Insubria.

Additional Information and Declarations

Competing Interests

Author Contributions

Animal Ethics

Data Availability

The authors declare that they have no competing interests.

Michela Bistoletti conceived and designed the experiments, performed the experiments, analyzed the data, prepared figures and/or tables, and approved the final draft.

Giovanni Micheloni conceived and designed the experiments, performed the experiments, analyzed the data, prepared figures and/or tables, and approved the final draft.

Nicolò Baranzini performed the experiments, analyzed the data, prepared figures and/or tables, and approved the final draft.

Annalisa Bosi performed the experiments, analyzed the data, prepared figures and/or tables, and approved the final draft.

Andrea Conti performed the experiments, analyzed the data, prepared figures and/or tables, and approved the final draft.

Viviana Filpa performed the experiments, analyzed the data, prepared figures and/or tables, and approved the final draft.

Cristina Pirrone performed the experiments, analyzed the data, prepared figures and/or tables, and approved the final draft.

Giorgia Millefanti performed the experiments, prepared figures and/or tables, and approved the final draft.

Elisabetta Moro performed the experiments, analyzed the data, prepared figures and/or tables, and approved the final draft.

Annalisa Grimaldi conceived and designed the experiments, analyzed the data, prepared figures and/or tables, authored or reviewed drafts of the paper, and approved the final draft.

Roberto Valli conceived and designed the experiments, authored or reviewed drafts of the paper, and approved the final draft.

Andreina Baj conceived and designed the experiments, analyzed the data, authored or reviewed drafts of the paper, and approved the final draft.

Francesca Crema conceived and designed the experiments, authored or reviewed drafts of the paper, and approved the final draft.

Cristina Giaroni conceived and designed the experiments, analyzed the data, prepared figures and/or tables, authored or reviewed drafts of the paper, and approved the final draft.

Giovanni Porta conceived and designed the experiments, authored or reviewed drafts of the paper, and approved the final draft.

The following information was supplied relating to ethical approvals (i.e., approving body and any reference numbers):

The protocol was approved by the Animal Care and Use Ethics Committee of the University of Pavia (Approval number n. 3/2011).

The following information was supplied regarding data availability:

Raw data and images of representative western blot are available in the Supplemental Files.

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
