# Peer review of "Homeoprotein OTX1 and OTX2 involvement in rat myenteric neuron adaptation after DNBS-induced colitis"

_PeerJ, doi:10.7717/peerj.8442_

## Round 0.1 · original submission · Minor Revisions

Please, address all the minor and major points raised by the three reviewers.

Reviewer 1 ·

Basic reporting

In the introduction section the nomenclature for chromosomal positions is not correct. Please Change the description for OTX1 and OTX2 into 2p15 and 14q22, respectively.

In the result section the text would gain understandability if each chapter ends with a short summary.

Experimental design

no comment

Validity of the findings

no comment

Additional comments

The paper submitted by Bistoletti and coworkers deals with the role of homeodomain proteins OTX1 and OTX2 in DNBS-induced colitis in rats. The study is well performed and written.

Reviewer 2 ·

Basic reporting

The manuscript is well structured. The findings are interesting, however, it is missing important information related to the modulations observed and mechanisms.
- Abstract: Ok
- Introduction: The introduction could be improved. Please include more previous date about OTX1 and OTX2
- methods: OK
- Results: See 2. experimental design.
- Discussion: There are a lot of introuction in the discussion. In the discussion section you must compare your date with other manuscripts from other groups.
- Conclusion: Please include only the conclusion, exclude speculation.

Experimental design

About the experiments;
1) It is missing bright field microscopy in all microscope images. Please include in
- Figure 1
- Figure 2
- Figure 5
- Figure 6
- Figure 7
- Figure 8
2) For OTX1, the staining is present in all cells and usually the co-locacization is always observed. Is it right?
3) The image of western-blot for OTX1 and OTX2 is not clear as described in the figures 9B and D.
4) The citokines must be quantified by elisa.
5) Please, include IL-17, IL-23 and TGF-beta. Colite is associated to TH17 response and Th17 was not evaluated.
6) Please correct, in mRNA you detect pro-IL1beta.

Validity of the findings

The authors used only one methodology to prove the finding (microscope). If used other techniques as Flow-cytometer will increase the strength of the manuscript.

Additional comments

Dear authors.
The manuscript is interesting, however, it is missing mechanisms.
1) What is the function of the increased of OTX1 and OTX2 after DNBS treatment?
2) What is the function of OTX1 superimposable with inducible nitric oxide synthase (iNos)? and nNOS?
3) Is it the increase of OTX1 and OTX2 associated to increase of citokines? Which?
4) Is the increase of OTX associated to microbiota? It will be interesting to treat mice with antibiotics.
5) If you prime cells in vitro with LPS, is it possible to induce the same mechanism?

Reviewer 3 ·

Basic reporting

no comment

Experimental design

no comment

Validity of the findings

no comment

Additional comments

The paper is interesting
I suggest the authors to add in the introduction other drugs used for intestinal inflammatios, see these papers:

Jo A, Yoo HJ, Lee M. Robustaflavone Isolated from Nandina domestica Using
Bioactivity-Guided Fractionation Downregulates Inflammatory Mediators. Molecules.
2019 May 8;24(9).

Szebeni GJ, Nagy LI, Berkó A, Hoffmann A, Fehér LZ, Bagyánszki M, Kari B,
Balog JA, Hackler L Jr, Kanizsai I, Pósa A, Varga C, Puskás LG. The
Anti-Inflammatory Role of Mannich Curcuminoids; Special Focus on Colitis.
Molecules. 2019 Apr 19;24(8).

Pagano E, Romano B, Iannotti FA, Parisi OA, D'Armiento M, Pignatiello S,
Coretti L, Lucafò M, Venneri T, Stocco G, Lembo F, Orlando P, Capasso R, Di Marzo
V, Izzo AA, Borrelli F. The non-euphoric phytocannabinoid cannabidivarin
counteracts intestinal inflammation in mice and cytokine expression in biopsies
from UC pediatric patients. Pharmacol Res. 2019 Nov;149:104464.


Pagano E, Capasso R, Piscitelli F, Romano B, Parisi OA, Finizio S, Lauritano A, Marzo VD, Izzo AA, Borrelli F. An Orally Active Cannabis Extract with High Content in Cannabidiol attenuates Chemically-induced Intestinal Inflammation and Hypermotility in the Mouse. Front Pharmacol. 2016 Oct 4;7:341.


Capasso R, Orlando P, Pagano E, Aveta T, Buono L, Borrelli F, Di Marzo V, Izzo AA. Palmitoylethanolamide normalizes intestinal motility in a model of post-inflammatory accelerated transit: involvement of CB₁ receptors and TRPV1 channels. Br J Pharmacol. 2014 Sep;171(17):4026-37.


The manuscript would benefit from inclusion of introducing/bridging sentences between the individual parts of the "Results" that explain the logical order and rationale for the experiments


In the Discussion, the Authors should highlight the possible clinical significance of their findings

---

## Round 0.2 · accepted · Accept

All the reviewer's suggestions were properly considered by the authors.